# Amortized Bayesian Meta-Learning

**Sachin Ravi and Alex Beatson**
Princeton University
{sachinr,abeatson}@cs.princeton.edu

## Abstract

Meta-learning, or learning-to-learn, has proven to be a successful strategy in attacking problems in supervised learning and reinforcement learning that involve small amounts of data. State-of-the-art solutions involve learning an initialization and/or optimization algorithm using a set of training episodes so that the meta-learner can generalize to an evaluation episode quickly. These methods perform well but often lack good quantification of uncertainty, which can be vital to real-world applications when data is lacking. We propose a meta-learning method which efficiently amortizes hierarchical variational inference across tasks, learning a prior distribution over neural network weights so that a few steps of Bayes by Backprop will produce a good task-specific approximate posterior. We show that our method produces good uncertainty estimates on contextual bandit and few-shot learning benchmarks.

## 1 Introduction

Deep learning has achieved success in domains that involve a large amount of labeled data (Oord et al., 2016; Huang et al., 2017) or training samples (Mnih et al., 2013; Silver et al., 2017). However, a key aspect of human intelligence is our ability to learn new concepts from only a few experiences. It has been hypothesized that this skill arises from accumulating prior knowledge and using it appropriately in new settings (Lake et al., 2017).

Meta learning attempts to endow machine learning models with the same ability by training a meta-learner to perform well on a distribution of training tasks. The meta-learner is then applied to an unseen task, usually assumed to be drawn from a task distribution similar to the one used for training, with the hope that it can learn to solve the new task efficiently. Popular meta-learning methods have advanced the state-of-the-art in many tasks, including the few-shot learning problem, where the model has to learn a new task given a small training set containing as few as one example per class. Though performance on few-shot learning benchmarks has greatly increased in the past few years, it is unclear how well the associated methods would perform in real-world settings, where the relationship between training and evaluation tasks could be tenuous. For success in the wild, in addition to good predictive accuracy, it is also important for meta-learning models to have good predictive uncertainty - to express high confidence when a prediction is likely to be correct but display low confidence when a prediction could be unreliable. This type of guarantee in predictive ability would allow appropriate human intervention when a prediction is known to have high uncertainty.

Bayesian methods offer a principled framework to reason about uncertainty, and approximate Bayesian methods have been used to provide deep learning models with accurate predictive uncertainty (Gal & Ghahramani, 2016; Louizos & Welling, 2017). By inferring a posterior distribution over neural network weights, we can produce a posterior predictive distribution that properly indicates the level of confidence on new unseen examples. Accordingly, we consider meta-learning under a Bayesian view in order to transfer the aforementioned benefits to our setting. Specifically, we extend the work of Amit & Meir (2018), who considered hierarchical variational inference for meta-learning. The work primarily dealt with PAC-Bayes bounds in meta-learning and the experiments consisted of data with tens of training episodes and small networks. In this paper, we show how the meta-learning framework of Finn et al. (2017) can be used to efficiently amortize variational inference for the Bayesian model of Amit & Meir (2018) in order to combine the former's flexibility and scalability with the latter's uncertainty quantification.

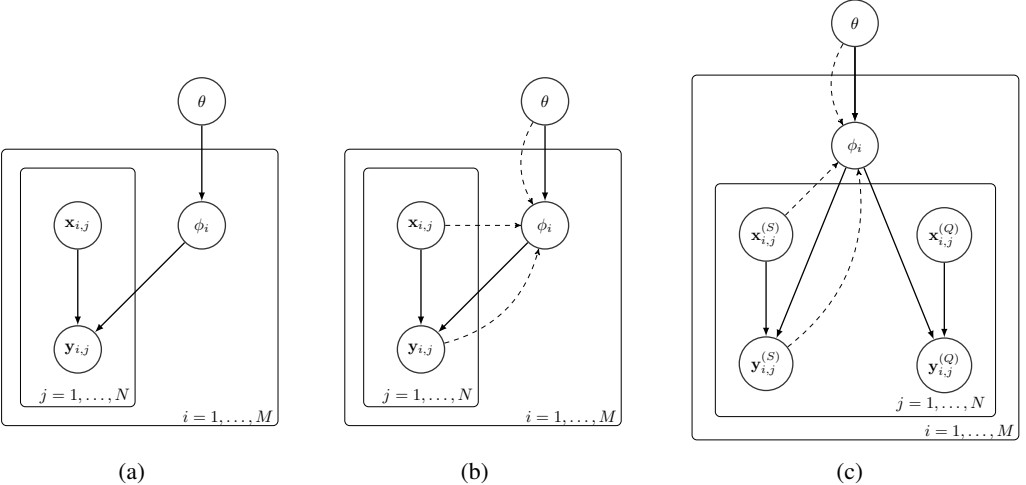

(a)          (b)          (c)

Figure 1: Graphical models for meta-learning framework. Dotted lines denote variational approximations. (a) Original setup in Amit & Meir (2018) where inference parameters are learned separately for each episode (b) Proposed initial amortized variational inference scheme (c) Proposed amortized variational inference scheme with support & query splits.

## 2   META-LEARNING VIA HIERARCHICAL VARIATIONAL INFERENCE

We first start by reviewing the hierarchical variational bayes formulation used in Amit & Meir (2018) for meta-learning. Assume we observe data from $M$ episodes, where the $i^{th}$ episode consists of data $\mathcal{D}_i$ containing $N$ data items, meaning $\mathcal{D}_i = \{(\mathbf{x}_{i,j}, \mathbf{y}_{i,j})\}_{j=1}^N$. We assume a hierarchical model with global latent variable $\theta$ and episode-specific variables $\phi_i$, $i = 1, \ldots M$ (see Figure 1).

Hierarchical variational inference can then be used to lower bound the likelihood of the data:

$$
\log \left[ \prod_{i=1}^M p(\mathcal{D}_i) \right] = \log \left[ \int p(\theta) \left[ \prod_{i=1}^M \int p(\mathcal{D}_i|\phi_i) p(\phi_i|\theta) \, d\phi_i \right] d\theta \right]
$$

$$
\geq \mathbb{E}_{q(\theta;\psi)} \left[ \log \left( \prod_{i=1}^M \int p(\mathcal{D}_i|\phi_i) p(\phi_i|\theta) \, d\phi_i \right) \right] - \mathrm{KL}(q(\theta;\psi)\|p(\theta))
$$

$$
= \mathbb{E}_{q(\theta;\psi)} \left[ \sum_{i=1}^M \log \left( \int p(\mathcal{D}_i|\phi_i) p(\phi_i|\theta) \, d\phi_i \right) \right] - \mathrm{KL}(q(\theta;\psi)\|p(\theta))
$$

$$
\geq \mathbb{E}_{q(\theta;\psi)} \left[ \sum_{i=1}^M \mathbb{E}_{q(\phi_i;\lambda_i)} \left[ \log p(\mathcal{D}_i|\phi_i) \right] - \mathrm{KL}(q(\phi_i;\lambda_i)\|p(\phi_i|\theta)) \right] - \mathrm{KL}(q(\theta;\psi)\|p(\theta))
$$

$$
= \mathcal{L}(\psi, \lambda_1, \ldots, \lambda_M).
$$

Here, $\psi$ and $\lambda_1, \ldots, \lambda_M$ are the variational parameters of the approximate posteriors over the global latent variables $\theta$ and the local latent variables $\phi_1, \ldots, \phi_M$, respectively.

Thus, variational inference involves solving the following optimization problem:

$$
\underset{\psi, \lambda_1 \ldots, \lambda_M}{\arg\max} \, \mathbb{E}_{q(\theta;\psi)} \left[ \sum_{i=1}^M \mathbb{E}_{q(\phi_i;\lambda_i)} \left[ \log p(\mathcal{D}_i|\phi_i) \right] - \mathrm{KL}(q(\phi_i;\lambda_i)\|p(\phi_i|\theta)) \right] - \mathrm{KL}(q(\theta;\psi)\|p(\theta))
$$
(1)

$$
\equiv \underset{\psi, \lambda_1 \ldots, \lambda_M}{\arg\min} \, \mathbb{E}_{q(\theta;\psi)} \left[ \sum_{i=1}^M -\mathbb{E}_{q(\phi_i;\lambda_i)} \left[ \log p(\mathcal{D}_i|\phi_i) \right] + \mathrm{KL}(q(\phi_i;\lambda_i)\|p(\phi_i|\theta)) \right] + \mathrm{KL}(q(\theta;\psi)\|p(\theta))
$$
(2)

Amit & Meir (2018) solve this optimization problem via mini-batch gradient descent on the objective starting from random initialization for all variational parameters. They maintain distinct variational parameters $\lambda_i$ for each episode $i$, each of which indexes a distribution over episode-specific weights $q(\phi_i; \lambda_i)$. While they only consider problems with at most 10 or so training episodes and where each $\phi_i$ is small (the weights of a 2-layer convolutional network), this approach is not scalable to problems with large numbers of episodes - such as few-shot learning, where we can generate millions of episodes by randomizing over classes and examples - and requiring deep networks.

## 3 Amortized Bayesian Meta-Learning

### 3.1 Scaling meta-learning with Amortized Variational Inference

Learning local variational parameters $\lambda_i$ for a large number of episodes $M$ becomes difficult as $M$ grows due to the costs of storing and computing each $\lambda_i$. These problems are compounded when each $\phi_i$ is the weight of a deep neural network and each $\lambda_i$ are variational parameters of the weight distribution (such as a mean and standard deviation of each weight). Instead of maintaining $M$ different variational parameters $\lambda_i$ indexing distributions over neural network weights $\phi_i$, we compute $\lambda_i$ on the fly with amortized variational inference (AVI), where a global learned model is used to predict $\lambda_i$ from $\mathcal{D}_i$. A popular use of AVI is training a variational autoencoder (Kingma & Welling, 2013), where a trained encoder network produces the variational parameters for each data point. Rather than training an encoder to predict $\lambda_i$ given the episode, we show that inference can be amortized by finding a good initialization, a la MAML (Finn et al., 2017). We represent the variational parameters for each episode as the output of several steps of gradient descent from a global initialization.

Let $\mathcal{L}_{\mathcal{D}_i}(\lambda, \theta) = -\mathbb{E}_{q(\phi_i; \lambda)}[\log p(\mathcal{D}_i|\phi_i)] + \mathrm{KL}(q(\phi_i; \lambda)\|p(\phi_i|\theta))$ be the part of the objective corresponding to data $\mathcal{D}_i$. Let the procedure $SGD_K(\mathcal{D}, \lambda^{(init)}, \theta)$ represent the variational parameters produced after $K$ steps of gradient descent on the objective $\mathcal{L}_{\mathcal{D}}(\lambda, \theta)$ with respect to $\lambda$ starting at the initialization $\lambda^{(0)} = \lambda^{(init)}$ and where $\theta$ is held constant i.e.:

1. $\lambda^{(0)} = \lambda^{(init)}$
2. for $k = 0, \ldots, K - 1$, set
   $\lambda^{(k+1)} = \lambda^{(k)} - \alpha\nabla_{\lambda^{(k)}}\mathcal{L}_{\mathcal{D}}(\lambda^{(k)}, \theta)$

We represent the variational distribution for each dataset $q_\theta(\phi_i|D_i)$ in terms of the local variational parameters $\lambda_i$ produced after $K$ steps of gradient descent on the loss for dataset $D_i$, starting from the global initialization $\theta$:

$$q_\theta(\phi_i|\mathcal{D}_i) = q(\phi_i; SGD_K(\mathcal{D}_i, \theta, \theta)).$$

Note that $\theta$ here serves as both the global initialization of local variational parameters and the parameters of the prior $p(\phi\,|\,\theta)$. We could pick a separate prior and global initialization, but we found tying the prior and initialization did not seem to have a negative affect on performance, while significantly reducing the number of total parameters necessary. With this form of the variational distribution, this turns the optimization problem of (2) into

$$\underset{\psi}{\arg\min}\, \mathbb{E}_{q(\theta;\psi)}\left[\sum_{i=1}^{M} -\mathbb{E}_{q_\theta(\phi_i|\mathcal{D}_i)}[\log p(\mathcal{D}_i|\phi_i)] + \mathrm{KL}(q_\theta(\phi_i|\mathcal{D}_i)\|p(\phi_i|\theta))\right] + \mathrm{KL}(q(\theta;\psi)\|p(\theta)).$$

$$(3)$$

Because each $q_\theta(\phi_i|D_i)$ depends on $\psi$ via $\theta$ (the initialization for the variational parameters before performing $K$ steps of gradient descent), we can also backpropagate through the computation of $q$ via the gradient descent process to compute updates for $\psi$. Though this backpropagation step requires computing the Hessian, it can be done efficiently with fast Hessian-vector products, which have been used in past work involving backpropagation through gradient updates (Maclaurin et al., 2015; Kim et al., 2018b). This corresponds to learning a global initialization of the variational parameters such that a few steps of gradient descent will produce a good local variational distribution for any given dataset.

We assume a setting where $M >> N$, i.e. we have many more episodes than data points within each episode. Accordingly, we are most interested in quantifying uncertainty within a given episode and

desire accurate predictive uncertainty in $q_\theta(\phi_i|D_i)$. We assume that uncertainty in the global latent variables $\theta$ should be low due to the large number of episodes, and therefore use a point estimate for the global latent variables, letting $q(\theta; \psi)$ be a dirac delta function $q(\theta) = \mathbb{1}\{\theta = \theta^*\}$. This removes the need for global variational parameters $\psi$ and simplifies our optimization problem to

$$\arg\min_\theta \left[ \sum_{i=1}^M -\mathbb{E}_{q_\theta(\phi_i|\mathcal{D}_i)} [\log p(\mathcal{D}_i|\phi_i)] + \text{KL}(q_\theta(\phi_i|\mathcal{D}_i)\|p(\phi_i|\theta)) \right] + \text{KL}(q(\theta)\|p(\theta)), \quad (4)$$

where $\theta^*$ is the solution to the above optimization problem. Note that $\text{KL}(q(\theta)\|p(\theta))$ term can be computed even when $q(\theta) = \mathbb{1}\{\theta = \theta^*\}$, as $\text{KL}(q(\theta)\|p(\theta)) = -\log p(\theta^*)$.

## 3.2 Amortized Variational Inference using only support set

In the few-shot learning problem, we must consider train and test splits for each dataset in each episode. Using notation from previous work on few-shot learning Snell et al. (2017), we will call the training examples in each dataset the *support set* and the test examples in each dataset the *query set* . Thus, $\mathcal{D}_i = \mathcal{D}_i^{(S)} \cup \mathcal{D}_i^{(Q)}$, where $\mathcal{D}_i^{(S)} = \left\{ (\mathbf{x}_{i,j}^{(S)}, \mathbf{y}_{i,j}^{(S)}) \right\}_{j=1}^N$ and $\mathcal{D}_i^{(Q)} = \left\{ (\mathbf{x}_{i,j}^{(Q)}, \mathbf{y}_{i,j}^{(Q)}) \right\}_{j=1}^{N'}$, and the assumption is that during evaluation, we are only given $\mathcal{D}_i^{(S)}$ to determine our variational distribution $q(\phi_i)$ and measure the performance of the model by evaluating the variational distribution on corresponding $\mathcal{D}_i^{(Q)}$. In order to match what is done during training and evaluation, we consider a modified version of the objective of (4) that incorporates this support and query set split. This means that for each episode $i$, we only have access to data $\mathcal{D}_i^{(S)}$ to compute the variational distribution, giving us the following objective:

$$\arg\min_\theta \left[ \sum_{i=1}^M -\mathbb{E}_{q_\theta\left(\phi_i\middle|\mathcal{D}_i^{(S)}\right)} [\log p(\mathcal{D}_i|\phi_i)] + \text{KL}\left( q_\theta\left(\phi_i\middle|\mathcal{D}_i^{(S)}\right) \middle\| p(\phi_i|\theta) \right) \right] + \text{KL}(q(\theta)\|p(\theta)),$$
$$(5)$$

where $q_\theta\left(\phi_i\middle|\mathcal{D}_i^{(S)}\right) = q\left(\phi_i; SGD_K\left(\mathcal{D}_i^{(S)}, \theta, \theta\right)\right)$. Note that the objective in this optimization problem still serves as a lower bound to the likelihood of all the episodic data because all that has changed is that we condition the variational distribution $q$ on less information (using only the support set vs using the entire dataset). Conditioning on less information potentially gives us a weaker lower bound for all the training datasets, but we found empirically that the performance during evaluation was better using this type of conditioning since there is no mismatch between how the variational distribution is computed during training vs evaluation.

## 3.3 Application Details

With the objective (5) in mind, we now give details on how we implement the specific model. We begin with the distributional forms of the priors and posteriors. The formulation given above is flexible but we consider fully factorized Gaussian distributions for ease of implementation and experimentation. We let $\theta = \{\boldsymbol{\mu_\theta}, \boldsymbol{\sigma_\theta^2}\}$, where $\boldsymbol{\mu_\theta} \in \mathbb{R}^D$ and $\boldsymbol{\sigma_\theta^2} \in \mathbb{R}^D$ represent the mean and variance for each neural network weight, respectively. Then, $p(\phi_i|\theta)$ is

$$p(\phi_i|\theta) = \mathcal{N}(\phi_i; \boldsymbol{\mu_\theta}, \boldsymbol{\sigma_\theta^2}\mathbf{I}),$$

and $q_\theta\left(\phi_i\middle|\mathcal{D}_i^{(S)}\right)$ is the following:

$$\left\{ \boldsymbol{\mu_\lambda^{(K)}}, \boldsymbol{\sigma^2}_{\boldsymbol{\lambda}}^{(K)} \right\} = SGD_K\left(\mathcal{D}_i^{(S)}, \theta, \theta\right)$$
$$q_\theta\left(\phi_i\middle|\mathcal{D}_i^{(S)}\right) = \mathcal{N}\left(\phi_i; \boldsymbol{\mu_\lambda^{(K)}}, \boldsymbol{\sigma^2}_{\boldsymbol{\lambda}}^{(K)}\right).$$

We let the prior $p(\theta)$ be

$$p(\theta) = \mathcal{N}(\boldsymbol{\mu}; \mathbf{0}, \mathbf{I}) \cdot \prod_{l=1}^D \text{Gamma}(\tau_l; a_0, b_0),$$

where $\tau_l = \frac{1}{\sigma_l^2}$ is the precision and $a_0$ and $b_0$ are the alpha and beta parameters for the gamma distribution. Note that with the defined distributions, the $SGD$ process here corresponds to performing Bayes by Backprop (Blundell et al., 2015) with the learned prior $p(\phi_i|\theta)$.

Optimization of (5) is done via mini-batch gradient descent, where we average gradients over multiple episodes at a time. The pseudo-code for training and evaluation is given in Algorithms 1 and 2 in the appendix. The KL-divergence terms are calculated analytically whereas the expectations are approximated by averaging over a number of samples from the approximate posterior, as has been done in previous work (Kingma & Welling, 2013; Blundell et al., 2015). The gradient computed for this approximation naively can have high variance, which can significantly harm the convergence of gradient descent (Kingma et al., 2015). Variance reduction is particularly important to the performance of our model as we perform stochastic optimization to obtain the posterior $q_\theta\left(\phi|D^{(S)}\right)$ at evaluation-time also. Previous work has explored reducing the variance of gradients involving stochastic neural networks, and we found this crucial to training the networks we use. Firstly, we use the Local Reparametrization Trick (LRT) (Kingma et al., 2015) for fully-connected layers and Flipout (Wen et al., 2018) for convolutional layers to generate fully-independent (or close to fully-independent in the case of Flipout) weight samples for each example. Secondly, we can easily generate multiple weight samples in the few-shot learning setting simply by replicating the data in each episode since we only have a few examples per class making up each episode. Both LRT and Flipout increase the operations required in the forward pass by 2 because they require two weight multiplications (or convolutions) rather than one for a normal fully-connected or convolutional layer. Replicating the data does not increase the run time too much because the total replicated data still fits on a forward pass on the GPU.

## 4 RELATED WORK

Meta-learning literature commonly considers the meta-learning problem as either empirical risk minimization (ERM) or bayesian inference in a hierarchical graphical model. The ERM perspective involves directly optimizing a meta learner to minimize a loss across training datasets (Bengio et al.; Schmidhuber, 1993). Recently, this has been successfully applied in a variety of models for few-shot learning (Vinyals et al., 2016; Finn et al., 2017; Snell et al., 2017; Mishra et al., 2018). The other perspective casts meta-learning as bayesian inference in a hierarchical graphical model (Tenenbaum, 1999; Fei-Fei & Perona, 2005; Koller et al., 2009). This approach provides a principled framework to reason about uncertainty. However, hierarchical bayesian methods once lacked the ability to scale to complex models and large, high-dimensional datasets due to the computational costs of inference. Recent developments in variational inference (Kingma & Welling, 2013; Blundell et al., 2015) allow efficient approximate inference with complex models and large datasets. These have been used to scale bayesian meta-learning using a variety of approaches. Edwards & Storkey (2016) infer episode-specific latent variables which can be used as auxillary inputs for tasks such as classification. As mentioned before, Amit & Meir (2018) learn a prior on the weights of a neural network and separate variational posteriors for each task.

Our method is very closely related to Finn et al. (2017) and recent work proposing Bayesian variants of MAML. Grant et al. (2018) provided the first Bayesian variant of MAML using the Laplace approximation. In concurrent work to this paper, Kim et al. (2018a) and Finn et al. (2018) propose Bayesian variants of MAML with different approximate posteriors. Finn et al. (2018) approximate MAP inference of the task-specific weights $\phi_i$, and maintain uncertainty only in the global model $\theta$. Our paper, however, considers tasks in which it is important to quantify uncertainty in task-specific weights - such as contextual bandits and few-shot learning. Kim et al. (2018a) focus on uncertainty in task-specific weights, as we do. They use a point estimate for all layers except the final layer of a deep neural network, and use Stein Variational Gradient Descent to approximate the posterior over the weights in the final layer with an ensemble. This avoids placing Gaussian restrictions on the approximate posterior; however, the posterior's expressiveness is dependant on the number of particles in the ensemble, and memory and computation requirements scale linearly and quadratically in the size of the ensemble, respectively. The linear scaling requires one to share parameters across particles in order to scale to larger datasets.

Moreover, there has been other recent work on Bayesian methods for few-shot learning. Neural Processes achieve Gaussian Process-like uncertainty quantification with neural networks, while being

easy to train via gradient descent (Garnelo et al., 2018a;b). However, it has not been demonstrated whether these methods can be scaled to bigger benchmarks like *mini*ImageNet. Gordon et al. (2018) adapt Bayesian decision theory to formulate the use of an amortization network to output the variational distribution over weights for each few-shot dataset. Both Kim et al. (2018a) and Gordon et al. (2018) require one to specify the global parameters (those that are shared across all episodes and are point estimates) vs task-specific parameters (those that are specific to each episode and have a variational distribution over them). Our method, however, does not require this distinction a priori and can discover it based on the data itself. For example, in Figure 5, which shows the standard deviations of the learned prior, we see that many of the 1$^{\text{st}}$ layer convolutional kernels have standard deviations very close to 0, indicating that these weights are essentially shared because there will be a large penalty from the prior for deviating from them in any episode. Not needing to make this distinction makes it more straightforward to apply our model to new problems, like the contextual bandit task we consider.

## 5 EVALUATION

We evaluate our proposed model on experiments involving contextual bandits and involving measuring uncertainty in few-shot learning benchmarks. We compare our method primarily against MAML. Unlike our model, MAML is trained by maximum likelihood estimation of the query set given a fixed number of updates on the support set, causing it to often display overconfidence in the settings we consider. For few-shot learning, we additionally compare against Probabilistic MAML (Finn et al., 2018), a Bayesian version of MAML that maintains uncertainty only in the global parameters.

### 5.1 CONTEXTUAL BANDITS

The first problem we consider is a contextual bandit task, specifically in the form of the wheel bandit problem introduced in Riquelme et al. (2018). The contextual bandit task involves observing a context $X_t$ from time $t = 0, \ldots, n$ and requires the model to select, based on its internal state and $X_t$, one of the $k$ available actions. Based on the context and the action selected at each time step, a reward is generated. The goal of the model is to minimize the cumulative regret, the difference between the sum of rewards of the optimal policy and the model's policy.

The wheel bandit problem is a synthetic contextual bandit problem with a scalar hyperparameter that allows us to control the amount of exploration required to be successful at the problem. The setup is the following: we consider a unit circle in $\mathbb{R}^2$ split up into 5 areas determined by the hyperparameter $\delta$. At each time step, the agent is given a point $X = (x_1, x_2)$ inside the circle and has to determine which arm to select among $k = 5$ arms. For $\|X\| \leq \delta$ (the low-reward region), the optimal arm is $k = 1$, which gives reward $r \sim \mathcal{N}(1.2, 0.01^2)$. All other arms in this area give reward $r \sim \mathcal{N}(1, 0.01^2)$. For $\|X\| > \delta$, the optimal arm depends on which of the 4 high-reward regions $X$ is in. Each of the 4 regions has an assigned optimal arm that gives reward $r \sim \mathcal{N}(50, 0.01^2)$, whereas the other 3 arms will give $r \sim \mathcal{N}(1.0, 0.01^2)$ and arm $k = 1$ will always give $r \sim \mathcal{N}(1.2, 0.01^2)$. The difficulty of the problem increases with $\delta$, as it requires increasing amount of exploration to determine where the high-reward regions are located. We refer the reader to Riquelme et al. (2018) for visual examples of the problem.

Thompson Sampling (Thompson, 1933) is a classic approach to tackling the exploration-exploitation trade-off involved in bandit problems which requires a posterior distribution over reward functions. At each time step an action is chosen by sampling a model from the posterior and acting optimally with respect to the sampled reward function. The posterior distribution over reward functions is then updated based on the observed reward for the action. When the posterior initially has high variance because of lack of data, Thompson Sampling explores more and turns to exploitation only when the posterior distribution becomes more certain about the rewards. The work of Riquelme et al. (2018) compares using Thompson Sampling for different models that approximate the posterior over reward functions on a variety of contextual bandit problems, including the wheel bandit.

We use the setup described in Garnelo et al. (2018b) to apply meta-learning methods to the wheel bandit problem. Specifically, for meta-learning methods there is a pre-training phase in which training episodes consist of randomly generated data across $\delta$ values from wheel bandit task. Then, these methods are evaluated using Thompson sampling on problems defined by specific values of

| $\delta$ | 0.5 | 0.7 | 0.9 | 0.95 | 0.99 |
|---|---|---|---|---|---|
| **n = 80, 000** | | | | | |
| Uniform | $100 \pm_{0.08}$ | $100 \pm_{0.09}$ | $100 \pm_{0.25}$ | $100 \pm_{0.37}$ | $100 \pm_{0.78}$ |
| NeuralLinear | $0.95 \pm_{0.02}$ | $1.60 \pm_{0.03}$ | $4.65 \pm_{0.18}$ | $9.56 \pm_{0.36}$ | $49.63 \pm_{2.41}$ |
| | | | | | |
| MAML | $\mathbf{0.20} \pm_{0.002}$ | $0.34 \pm_{0.004}$ | $1.02 \pm_{0.01}$ | $2.10 \pm_{0.03}$ | $9.81 \pm_{0.27}$ |
| Our Model | $0.22 \pm_{0.002}$ | $\mathbf{0.29} \pm_{0.003}$ | $\mathbf{0.66} \pm_{0.008}$ | $\mathbf{1.03} \pm_{0.01}$ | $\mathbf{4.66} \pm_{0.10}$ |
| **n = 2, 000** | | | | | |
| Uniform | $100 \pm_{0.25}$ | $100 \pm_{0.42}$ | $100 \pm_{0.79}$ | $100 \pm_{1.15}$ | $100 \pm_{1.88}$ |
| MAML | $1.79 \pm_{0.04}$ | $2.10 \pm_{0.04}$ | $6.08 \pm_{0.47}$ | $16.80 \pm_{1.30}$ | $55.53 \pm_{2.18}$ |
| Our Model | $\mathbf{1.36} \pm_{0.03}$ | $\mathbf{1.59} \pm_{0.04}$ | $\mathbf{3.51} \pm_{0.17}$ | $\mathbf{7.21} \pm_{0.41}$ | $\mathbf{35.04} \pm_{1.93}$ |

Table 1: Cumulative regret results on the wheel bandit problem with varying $\delta$ values. Results are normalized with the performance of the uniform agent (as was done in Riquelme et al. (2018)) and results shown are mean and standard error for cumulative regret calculated across 50 trials

$\delta$. We can create a random training episode for pre-training by first sampling $M$ different wheel problems $\{\delta_i\}_{i=1}^{M}, \delta_i \sim \mathcal{U}(0, 1)$, followed by sampling tuples of the form $\{(X, a, r)\}_{j=1}^{N}$ for context $X$, action $a$, and observed reward $r$. As in Garnelo et al. (2018b), we use $M = 64$ and $N = 562$ (where the support set has $512$ items and the query set has $50$ items). We then evaluate the trained meta-learning models on specific instances of the wheel bandit problem (determined by setting the $\delta$ hyperparameter). Whereas the models in Riquelme et al. (2018) have no prior knowledge to start off with when being evaluated on each problem, meta-learning methods, like our model and MAML, have a chance to develop some sort of prior that they can utilize to get a head start. MAML learns a initialization of the neural network that it can then fine-tune to the given problem data, whereas our method develops a prior over the model parameters that can be utilized to develop an approximate posterior given the new data. Thus, we can straightforwardly apply Thompson sampling in our model using the approximate posterior at each time step whereas for MAML we just take a greedy action at each time step given the current model parameters.

The results of evaluating the meta-learning methods using code made available by authors of Riquelme et al. (2018) after the pre-training phase are shown in Table 1. We also show results from NeuralLinear, one of the best performing models from Riquelme et al. (2018), to display the benefit of the pre-training phase for the meta-learning methods. We vary the number of contexts and consider $n = 80, 000$ (which was used in Riquelme et al. (2018)) and $n = 2, 000$ (to see how the models perform under fewer time steps). We can see that as $\delta$ increases and more exploration is required to be successful at the problem, our model has a increasingly better cumulative regret when compared to MAML. Additionally, we notice that this improvement is even larger when considering smaller amount of time steps, indicating that our model converges to the optimal actions faster than MAML. Lastly, in order to highlight the difference between our method and MAML, we visualize the learned prior $p(\phi \,|\, \theta)$ in Figure 2 by showing the expectation and standard-deviation of predicted rewards for specific arms with respect to the prior. We can see that the standard deviation of the central low-reward arm is small everywhere, as there is reward little variability in this arm across $\delta$ values. For the high-reward arm in the upper-right corner, we see that the standard deviation is high at the edges of the area in which this arm can give high reward (depending on the sampled $\delta$ value). This variation is useful during exploration as this is the region in which we would like to target our exploration to figure out what $\delta$ value we are faced with in a new problem. MAML is only able to learn the information associated with expected reward values and so is not well-suited for appropriate exploration but can only be used in a greedy manner.

## 5.2 FEW-SHOT LEARNING

We consider two few-shot learning benchmarks: **CIFAR-100** and *mini***ImageNet**, where both datasets consist of 100 classes and 600 images per class and where CIFAR-100 has images of size $32 \times 32$ and *mini*ImageNet has images of size $84 \times 84$. We split the 100 classes into separate sets of 64 classes for training, 16 classes for validation, and 20 classes for testing for both of the datasets (using the split from Ravi & Larochelle (2016) for *mini*ImageNet, while using our own for CIFAR-100 as a commonly used split does not exist). For both benchmarks, we use the convolutional architecture

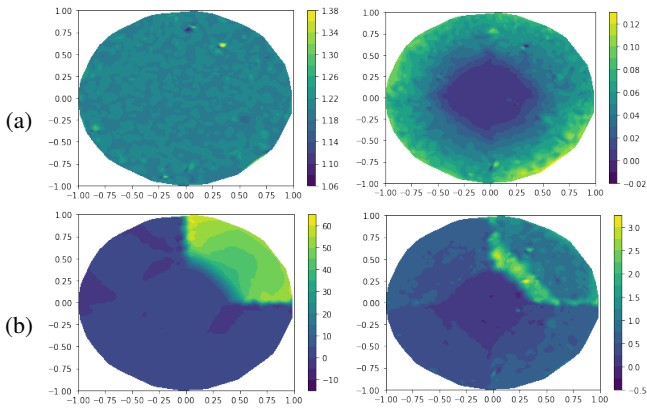

Figure 2: Visualization of arm rewards according to prior distribution of our model. (a) expectation and standard-deviation of low-reward arm (computed by sampling weights from the prior) evaluated on points on unit circle. (b) expectation and standard-deviation of one of the high-reward arms computed in same way as for low-reward arm.

| CIFAR-100 | 1-shot | |
|---|---|---|
| | 5-class | 10-class |
| MAML (ours) | $51.6 \pm 0.74$ | $36.2 \pm 0.46$ |
| Prob. MAML (ours) | $52.8 \pm 0.75$ | $36.6 \pm 0.44$ |
| Our Model | $49.5 \pm 0.74$ | $35.7 \pm 0.47$ |

| *mini*ImageNet | 1-shot, 5-class |
|---|---|
| MAML (ours) | $47.0 \pm 0.59$ |
| Prob. MAML (ours) | $47.8 \pm 0.61$ |
| Our Model | $45.0 \pm 0.60$ |

Table 2: Few-shot classification accuracies with 95% confidence intervals on CIFAR-100 and *mini*ImageNet.

used in Finn et al. (2017), which consists of 4 convolutional layers, each with 32 filters, and a fully-connected layer mapping to the number of classes on top. For the few-shot learning experiments, we found it necessary to downweight the inner KL term for better performance in our model.

While we focus on predictive uncertainty, we start by comparing classification accuracy of our model compared to MAML. We consider 1-shot, 5-class and 1-shot, 10-class classification on CIFAR-100 and 1-shot, 5-class classification on *mini*ImageNet, with results given in Table 2. For both datasets, we compare our model with our own re-implementation of MAML and Probabilistic MAML. Note that the accuracy and associated confidence interval for our implementations for *mini*ImageNet are smaller than the reference implementations because we use a bigger query set for test episodes (15 vs 1 example(s) per class) and we average across more test episodes (1000 vs 600), respectively, compared to Finn et al. (2017). Because we evaluate in a transductive setting (Nichol & Schulman, 2018), the evaluation performance is affected by the query set size, and we use 15 examples to be consistent with previous work (Ravi & Larochelle, 2016). Our model achieves comparable to a little worse on classification accuracy than MAML and Probabilistic MAML on the benchmarks.

To measure the predictive uncertainty of the models, we first compute reliability diagrams (Guo et al., 2017) across many different test episodes for both models. Reliability diagrams visually measure how well calibrated the predictions of a model are by plotting the expected accuracy as a function of the confidence of the model. A well-calibrated model will have its bars align more closely with the diagonal line, as it indicates that the probability associated with a predicted class label corresponds closely with how likely the prediction is to be correct. We also show the Expected Calibration Error (ECE) and Maximum Calibration Error (MCE) of all models, which are two quantitative ways to measure model calibration (Naeini et al., 2015; Guo et al., 2017). ECE is a weighted average of each bin's accuracy-to-confidence difference whereas MCE is the worst-case bin's accuracy-to-confidence difference. Reliability diagrams and associated error scores are shown in Figure 3. We see that across different tasks and datasets, the reliability diagrams and error scores reflect the fact that our model is always better calibrated on evaluation episodes compared to MAML and Probabilitic MAML.

Another way we can measure the quality of the predictive uncertainty of a model is by measuring its confidence on out-of-distribution examples from unseen classes. This tests the model's ability to be uncertain on examples it clearly does not know how to classify. One method to visually measure this is by plotting the empirical CDF of a model's entropies on these out-of-distribution examples (Louizos & Welling, 2017). A model represented by a CDF curve that is towards the bottom-right is preferred, as it indicates that the probability of observing a high confidence prediction from the

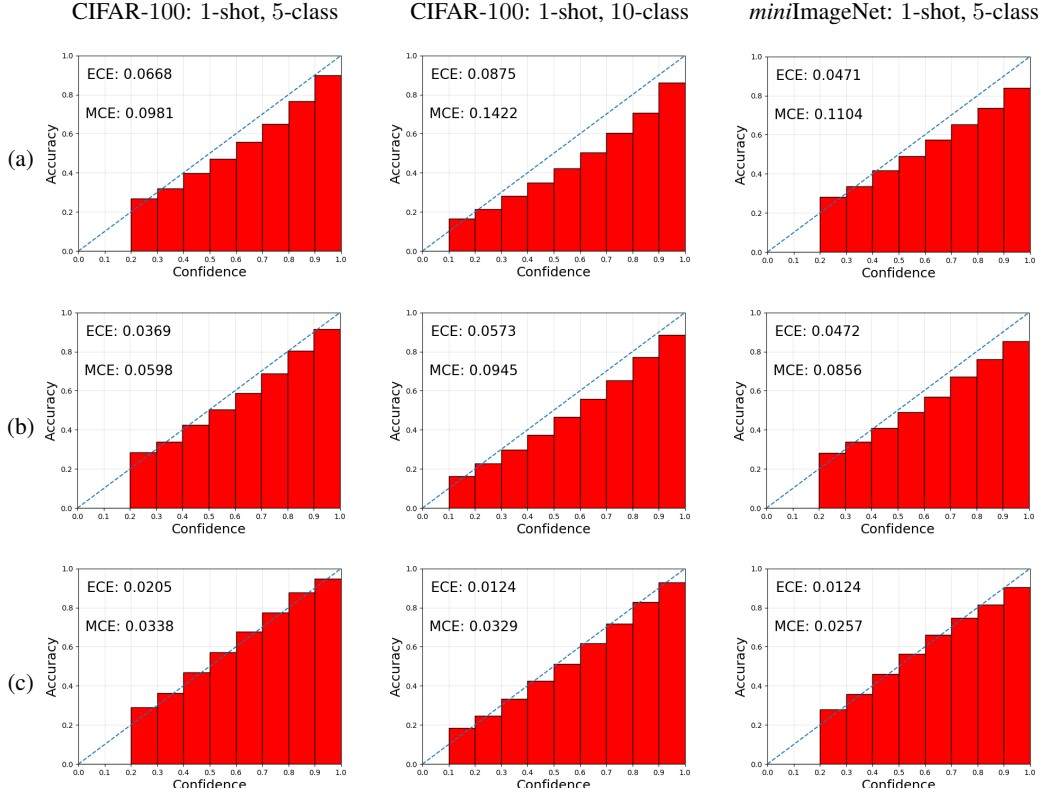

Figure 3: Reliability diagrams for MAML and our model on various tasks across datasets. Relibiality diagrams are computed by gathering predicted probabilities for query set examples across many episodes, where the same set of evaluation episodes are used for both models. (a) MAML (b) Probabilistic MAML (c) Our model.

model is low on an out-of-distribution example. We can plot the same type of curve in our setting by considering the model's confidence on out-of-episode examples for each test episode. Empirical CDF curves for both MAML-based models and our model are shown in Figure 4. We see that in general our model computes better uncertainty estimates than the comparison models, as the probability of a low entropy prediction is always smaller.

Lastly, we visualize the prior distribution $p(\phi|\theta)$ that has been learned in tasks involving deep convolutional networks. We show the standard deviations of randomly selected filters from the first convolutional layer to the last convolutional layer from our CIFAR-100 network trained on 1-shot, 5-class task in Figure 5. Interestingly, the standard deviation of the prior for the filters increases as we go higher up in the network. This pattern reflects the fact that across the training episodes the prior can be very confident about the lower-level filters, as they capture general, useful lower-level features and so do not need to be modified as much on a new episode. The standard deviation for the higher-level filters is higher, reflecting that fact that these filters need to be fine-tuned to the labels present in the new episode. This variation in the standard deviation represents different learning speeds across the network on a new episode, indicating which type of weights are general and which type of weights need to be quickly modified to capture new data.

## 6 CONCLUSION

We described a method to efficiently use hierarchical variational inference to learn a meta-learning model that is scalable across many training episodes and large networks. The method corresponds to learning a prior distribution over the network weights so that a few steps of Bayes by Backprop will produce a good approximate posterior. Through various experiments we show that using a Bayesian interpretation allows us to reason effectively about uncertainty in contextual bandit and

CIFAR-100: 1-shot, 5-class          CIFAR-100: 1-shot, 10-class

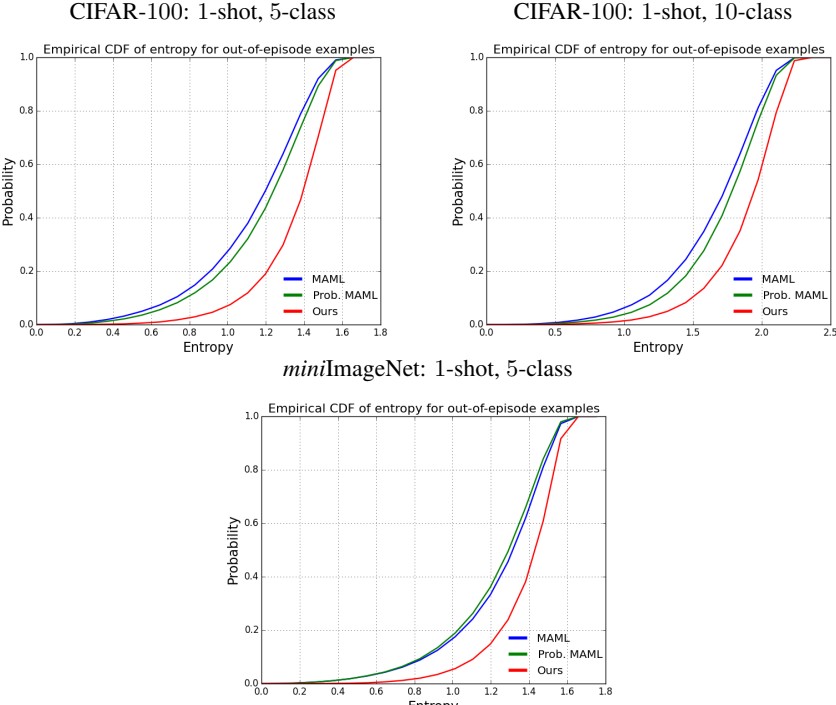

*mini*ImageNet: 1-shot, 5-class

Figure 4: Comparison of empirical CDF of entropy of predictive distributions on out-of-episode examples on various tasks and datasets. Data for CDF comes from computing the entropy on out-of-episode examples across many episodes, where out-of-episode examples are generated by randomly sampling classes not belonging to the episode and randomly sampling examples from those classes. The same set of evaluation episodes are used for both models.

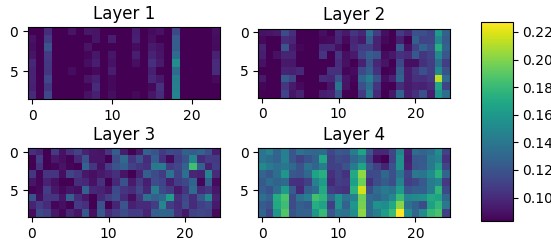

Figure 5: Standard deviation of prior for convolutional kernels across layers of network. The x-axis indexes different filters in each layer whereas the y-axis indexes across positions in the $3 \times 3$ kernel.

few-shot learning tasks. The proposed method is flexible and future work could involve considering more expressive prior (and corresponding posterior) distributions to further improve the uncertainty estimates.

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

## 7 APPENDIX

### 7.1 PSEUDOCODE

In algorithms 1 and 2 we give the pseudocode for meta-training and meta-evaluation, respectively. Note that in practice, we do not directly parameterize variance parameters but instead parameterize the standard deviation as the output of the softplus function as was done in Blundell et al. (2015) so that it is always non-negative.

---

**Algorithm 1** Meta-training

---

**Input**: Number of update steps $K$, Number of total episodes $M$, Inner learning rate $\alpha$, Outer learning rate $\beta$

1: Initialize $\theta = \left\{ \boldsymbol{\mu}_\theta, \boldsymbol{\sigma}_\theta^2 \right\}$
2: $p(\theta) = \mathcal{N}(\boldsymbol{\mu}; \mathbf{0}, \mathbf{I}) \cdot \prod_{l=1}^D \text{Gamma}(\tau_l; a_0, b_0)$
3: **for** $i = 1$ to $M$ **do**
4: $\quad \mathcal{D}_i = \left\{ \mathcal{D}_i^{(S)}, \mathcal{D}_i^{(Q)} \right\}$
5: $\quad \boldsymbol{\mu}_\lambda^{(0)} \leftarrow \boldsymbol{\mu}_\theta; \boldsymbol{\sigma^2}_\lambda^{(0)} \leftarrow \boldsymbol{\sigma}_\theta^2$
6: $\quad$ **for** $k = 0$ to $K - 1$ **do**
7: $\quad\quad \lambda^{(k)} \leftarrow \left\{ \boldsymbol{\mu}_\lambda^{(k)}, \boldsymbol{\sigma}_\lambda^{(k)} \right\}$
8: $\quad\quad \boldsymbol{\mu}_\lambda^{(k+1)} \leftarrow \boldsymbol{\mu}_\lambda^{(k)} - \alpha \nabla_{\boldsymbol{\mu}_\lambda^{(k)}} \mathcal{L}_{\mathcal{D}_i^{(S)}} \left( \lambda^{(k)}, \theta \right)$
9: $\quad\quad \boldsymbol{\sigma^2}_\lambda^{(k+1)} \leftarrow \boldsymbol{\sigma^2}_\lambda^{(k)} - \alpha \nabla_{\boldsymbol{\sigma^2}_\lambda^{(k)}} \mathcal{L}_{\mathcal{D}_i^{(S)}} \left( \lambda^{(k)}, \theta \right)$
10: $\quad$ **end for**
11:
12: $\quad \lambda^{(K)} \leftarrow \left\{ \boldsymbol{\mu}_\lambda^{(K)}, \boldsymbol{\sigma^2}_\lambda^{(K)} \right\}$
13: $\quad q(\theta) = \mathbb{1}\{\boldsymbol{\mu} = \boldsymbol{\mu}_\theta\} \cdot \mathbb{1}\{\boldsymbol{\sigma}^2 = \boldsymbol{\sigma}_\theta^2\}$
14: $\quad \boldsymbol{\mu}_\theta \leftarrow \boldsymbol{\mu}_\theta - \beta \nabla_{\boldsymbol{\mu}_\theta} \left[ \mathcal{L}_{\mathcal{D}_i}(\lambda^{(K)}, \theta) + \frac{1}{M} \text{KL}(q(\theta) \| p(\theta)) \right]$
15: $\quad \boldsymbol{\sigma}_\theta^2 \leftarrow \boldsymbol{\sigma}_\theta^2 - \beta \nabla_{\boldsymbol{\sigma}_\theta^2} \left[ \mathcal{L}_{\mathcal{D}_i}(\lambda^{(K)}, \theta) + \frac{1}{M} \text{KL}(q(\theta) \| p(\theta)) \right]$
16: **end for**

---

**Algorithm 2** Meta-evaluation

---

**Input**: Number of update steps $K$, Dataset $\mathcal{D} = \left\{ \mathcal{D}^{(S)}, \mathcal{D}^{(Q)} \right\}$, Parameters $\theta = \left\{ \boldsymbol{\mu}_\theta, \boldsymbol{\sigma}_\theta^2 \right\}$, Inner learning rate $\alpha$

1: $\boldsymbol{\mu}_\lambda^{(0)} \leftarrow \boldsymbol{\mu}_\theta; \boldsymbol{\sigma^2}_\lambda^{(0)} \leftarrow \boldsymbol{\sigma}_\theta^2$
2: **for** $k = 0$ to $K - 1$ **do**
3: $\quad \lambda^{(k)} \leftarrow \left\{ \boldsymbol{\mu}_\lambda^{(k)}, \boldsymbol{\sigma}_\lambda^{(k)} \right\}$
4: $\quad \boldsymbol{\mu}_\lambda^{(k+1)} \leftarrow \boldsymbol{\mu}_\lambda^{(k)} - \alpha \nabla_{\boldsymbol{\mu}_\lambda^{(k)}} \mathcal{L}_{\mathcal{D}^{(S)}} \left( \lambda^{(k)}, \theta \right)$
5: $\quad \boldsymbol{\sigma^2}_\lambda^{(k+1)} \leftarrow \boldsymbol{\sigma^2}_\lambda^{(k)} - \alpha \nabla_{\boldsymbol{\sigma^2}_\lambda^{(k)}} \mathcal{L}_{\mathcal{D}^{(S)}} \left( \lambda^{(k)}, \theta \right)$
6: **end for**
7:
8: $q_\theta \left( \phi \,\middle|\, D^{(S)} \right) = \mathcal{N} \left( \phi; \boldsymbol{\mu}_\lambda^{(K)}, \boldsymbol{\sigma^2}_\lambda^{(K)} \right)$
9: Evaluate $D^{(Q)}$ using $\mathbb{E}_{q_\theta\left(\phi \,\middle|\, D^{(S)}\right)} \left[ p(D^{(Q)} \,\middle|\, \phi) \right]$

---

## 7.2 HYPERPARAMETERS

### 7.2.1 CONTEXTUAL BANDITS

|  | $n = 2,000$ | $n = 80,000$ |
|---|---|---|
| Number of NN Layers | 2 | 2 |
| Hidden Units per Layer | 100 | 100 |
| $t_s$ (mini-batches per training step) | 100 | 100 |
| $t_f$ (frequency of training) | 20 | 100 |
| Optimizer | Adam | Adam |
| Learning rate | 0.001 | 0.001 |

Table 3: Hyperparameters for contextual bandit experiments. These hyperparameters were used for both MAML and our model when comparing them. Hyperparameters $t_f$ and $t_s$ were used as defined in Riquelme et al. (2018) .

### 7.2.2 FEW-SHOT LEARNING

|  | CIFAR-100 | *mini*ImageNet |
|---|---|---|
| Inner Learning Rate | 0.1 | 0.1 |
| Outer Learning Rate | 0.001 | 0.001 |
| Gradient steps for $q$ (training) | 5 | 5 |
| Gradient steps for $q$ (evaluation) | 10 | 10 |
| Number of samples to compute expectation for updating $q$ | 5 | 5 |
| Number of samples to compute expectation for outer-loss | 2 | 2 |
| Number of samples to compute expectation for evaluation | 10 | 10 |
| $a_0$ for hyper-prior | 2 | 1 |
| $b_0$ for hyper-prior | 0.2 | 0.01 |

Table 4: Hyperparameters for our model for few-shot learning experiments.

