# OpenReview forum: "Amortized Bayesian Meta-Learning"
_ICLR.cc/2019/Conference_

### Official Review · AnonReviewer2 · 2018-11-02
**Good paper. Unclear that it makes a significant contribution, either in terms of novelty or empirical results**

**Rating:** 6
**Confidence:** 3

**Review:**

This work proposes an adaptation to MAML-type models that accounts for posterior uncertainty in task specific latent variables. This is achieved via a hierarchical Bayesian view of MAML, employing variational inference for the task-specific parameters. The key intuition of this paper is that one can perform fast and efficient test-time variational inference for the task-specific latent variables by learning a good initialization during meta-training. This is achieved in a very similar fashion to MAML, and allows for an interesting form of amortization of test-time inference.

Pros:
- For the most part, the approach presented is principled and well justified.
- The motivation is clear: in the few-shot learning regime we expect to have little data to infer the task-specific latent variables, and so we should perform posterior inference to account for uncertainty.
- The paper is well written, clear, and easy to follow.

Cons (more details below):
- It is not clear what the significant contributions of this paper are, as a number of methods have been proposed to account for uncertainty in the task-specific latent variables, and results for many of these methods appear to be better than those presented here.
- Experimental section does compare to many of the existing related methods
- There are some conceptual issues that need to be addressed by the authors.

I enjoyed reading this paper, and I think the ideas and work presented are, for the most part, solid. However, I am not sure to what extent the novel contribution of this paper is significant. Several papers, including Grant et al. (2018), but going back to Heskes (2000), have proposed the hierarchical Bayesian view of meta-learning. Grant et al. (2018) used a Laplace approximation to learn in such a model with MAML-type settings, presenting a method that accounts for uncertainty in this family of models. More recently, Finn et al. (2018) and Kim et al. (2018) have done this in a variational manner, albeit with variations in the implementation details. Gordon et al. (2018) proposed a more general presentation, unifying the above works (and others) in a Bayesian framework that allows for different functional forms of posterior inference (both point estimates and distributional) of the task-specific parameters, including gradient based procedures. All of these papers have been publicly available for a few months at the time of submission, such that this view of meta-learning as (amortized) Bayesian inference is not novel.

Here are some points that I would ask the authors to address during the rebuttal period:

- The method presented in the paper does not account for the meta-training splits into query and test sets, other than to mention that these led to empirical performance gains (this is somewhat typical of probabilistic meta-learning papers). However, it not clear that this is justified from a probabilistic inference perspective, which would favour conditioning on all available data at inference time. Further, in the experimental section, the authors state that "For the few-shot learning experiments, we found it necessary to downweight the inner KL term for better performance in our model". Put together, it is not quite clear exactly what form of approximate inference is being conducted here. Can the authors comment on this?

- I am not sure I agree with the authors' interpretation of the term "amortized Bayesian inference", at least in that it deviates from the way the term is typically used in the related literature. The method negates the need to maintain variational parameters for each latent variable, and approximate posterior inference for unseen tasks may be performed relatively efficiently, which is highly desirable. However, a gradient optimization procedure must still be performed for inference of task-specific variables for new tasks at test time. Thus, new variational parameters must be introduced and optimized at test time. It is true that by finding good global initializations the authors may drastically reduce the computational cost of the inference process, but this implies that the cost of inference at test time has been reduced, not fully amortized to a fixed cost (unless one fixes the number of gradient steps, which is a further deviation from variational inference and requires a prior of the form used in Grant et al. (2018)). Full amortization of inference for the task-specific variables is proposed by Garnelo et al. (2018) and Gordon et al. (2018), as well as Edwards and Storkey (2016), all of which employ inference networks mapping directly from the query sets to the variational parameters of the latent variables. In these cases, posterior inference of the latent variables for unseen tasks has the constant cost of a pass through an inference network, rather than several forward-backward passes, and does not require introducing new variational parameters to be optimized. Further, these methods negate the need for differentiating through gradient-based procedures at meta-training time, which is not avoided in this paper, but rather dealt with in the standard Hessian-vector product form. It would be highly useful in the paper (perhaps in the related work section) for the authors to conduct a more thorough comparison of their proposed method and the existing literature employing amortized inference for meta-learning, to put their work in context.

I also have a number of concerns regarding the experimental section of the paper, which I find to be lacking both in details and the empirical comparison of the method to existing works.
- The authors' cite recent works on meta-learning that take into account uncertainty in the local latent variables (e.g., Grant et al. (2018), Finn et al. (2018), Kim et al. (2018)), but do not compare to these methods.
- Results from Garnelo et al. (2018) are not provided for the contextual bandits experiment. Their results seem to be comparable or better to those presented in this paper. Can the authors comment on this?
- The same is true for the few-shot learning case, where MAML is the only method compared to, despite there being, at the time of submission, many papers which have significantly improved upon these results.
- In terms of details, it is unclear how many gradient steps were taken at test time, and how this affects performance of the model.
- In terms of accuracy, the proposed method appears to be under-performing significantly (i.e., below confidence bounds in almost all cases).
- The statement "...we believe improvements could be made with better variance reduction methods for stochastic gradients" should, in my opinion, either be investigated or omitted from the paper.
- In terms of uncertainty quantification, I find this experimental evaluation highly interesting. However, there is not a comparison to much of the existing work. The comparison to MAML is only of moderate interest in this case, as MAML is a deterministic method and is not expected to perform well in this regard. A comparison to Probabilistic or Bayesian MAML (at the least) would be more convincing if uncertainty calibration proved to be better for this method.

Overall, the paper proposes a principled approach to performing approximate posterior inference for task specific latent variables in meta-learning settings. The paper is well-written, and the method is clearly derived. However, it is my impression that the paper does not make significant novel contributions to the existing research in (probabilistic) meta-learning, does not properly acknowledge all existing work (much of which covers the main ideas presented in the paper), has a number of conceptual issues that might need addressing, and its experimental section lacks evaluation and comparisons to the existing similar works. As the method is, for the most part, principled and well-derived, and the paper well written, I am willing to reconsider my overall score if the authors can demonstrate either (i) significant novelty or (ii) that this particular flavour of inference for the task-specific parameters provides significant benefits over existing approaches.

[1] - T. Heskes. Empirical Bayes for learning to learn. 2000.
[2] - E. Grant, C. Finn, S. Levine, T. Darrell, and T. Griffiths. Recasting gradient-based meta-learning as hierarchical Bayes. 2018.
[3] - C. Finn, K. Xu, and S. Levine. Probabilistic model-agnostic meta-learning. 2018.
[4] - T. Kim, J. Yoon, O. Dia, S. Kim, Y. Bengio, and S. Ahn. Bayesian model-agnostic meta-learning. 2018.
[5] - J. Gordon, J. Bronskill, M. Bauer, S. Nowozin, and R. Turner. Decision-theoretic meta-learning: versatile and efficient amortization of few-shot learning. 2018.
[6] - M. Garnelo, J. Schwarz, D. Rosenbaum, F. Viola, D. J. Rezende, S. Eslami, and Y. W. Teh. Neural processes. 2018.
[7] - H. Edwards, and A. Storkey. Towards a neural statistician. 2016.

---

> ### Author Response · Authors · 2018-11-20
> **Response to Reviewer 2 (1/2)**
>
> Thank you for your review and detailed feedback! We’ve addressed the points you talked about below.
>
> => “The method presented...does not account for the meta-training splits into query and test sets…”
>
> We are free to choose any parameterized procedure to produce an approximate posterior. Therefore, we can choose to condition on only the support set when computing the approximate posterior. It is true that conditioning on less information may give a looser lower bound whereas conditioning on all the data would make the bound tighter. However, we only care about obtaining a tight bound during training insofar as it allows computing a good approximate posterior from the support set at test time. Therefore, we choose to condition our approximate posterior on only the support set. We wish our model’s performance to generalize from training to test, and so we ensure the training and test conditions are similar. This is supported by empirical evidence - the model performs better during testing when the variational distribution is computed the same way during training and testing.
>
> Also one can view the objective in Eqn 5 (specifically the part inside the brackets corresponding to each episode i) as the KL between the approximate posterior conditioned on the support set and the true posterior over task-specific latent variables phi conditioned on the support & query sets and theta. This is similar to loss used in Bayesian MAML, which aims to minimize a dissimilarity function between an approximate posterior over task-specific parameters given the support set, and an approximate posterior over task-specific parameters given the support & query sets. The loss there is presented without derivation from a probabilistic inference perspective and justified by empirical performance. It’s interesting that our derived loss connects to theirs.
>
> => “Further...the authors state that ‘For the few-shot learning experiments, we found it necessary to downweight the inner KL term for better performance in our model’...”
>
> Downweighting the KL term can be justified from a probabilistic perspective as accounting for confidence in the data (controlled by label noise and # of examples). The weight on the KL term allows us to control the assumed noise in the observation model. We choose the weight on the KL term to maximize the validation performance.
>
> => “I am not sure I agree with the authors' interpretation of the term ‘amortized Bayesian inference’...”
>
> We agree that our method could be characterized as “semi-amortized”, a la Semi-Amortized VAEs (Kim et al). In practice, we do fix a small number of gradient steps, which effectively does mean that finding a task-specific posterior is a fixed cost.
> Kim, Yoon et al. Semi-Amortized Variational Autoencoders. 2018.
>
> => “It is true that by finding good global initializations the authors may drastically reduce the computational cost of the inference process, but this implies that the cost of inference at test time has been reduced, not fully amortized to a fixed cost (unless one fixes the number of gradient steps, which is a further deviation from variational inference and requires a prior of the form used in Grant et al.).”
>
> We disagree that the parameterized procedure we use (with fixed number of gradient steps) to produce an approximate posterior requires changing the structure of the prior. Automatic-differentiation based variational inference (for example, the VAE) is based on the idea that given a parametric differentiable procedure (such as the forward pass through an encoder network) that produces an approximate posterior, we can train the parameters of that procedure in order to to produce a better approximate posterior (through maximizing the ELBO). Importantly, this does not require changing the structure of the prior and is valid for any prior we define - as long as we can estimate the KL divergence. Using gradient descent with a fixed number of steps from initial variational weights can be thought of analogously to using an encoder network. It could be possible that our method suffers from an amortization gap (Cremer et al), as do encoder networks, because of using a fixed number of updates. However, we didn’t observe this in practice - we experimented with different amounts of steps and found diminishing returns after a certain point. Please do let us know if we have misunderstood what you meant and we’d be happy to discuss further.
> Cremer, Chris et al. Inference Suboptimality in Variational Autoencoders. 2018.
>
> => “Full amortization of inference for the task-specific variables is proposed by Garnelo et al and Gordon et al, as well as Edwards & Storkey, all of which employ inference networks mapping directly from the query sets to the variational parameters of the latent variables. In these cases, posterior inference of the latent variables for unseen tasks has the constant cost of a pass through an inference network, rather than several forward-backward passes...” [cont.]

---

> > ### Author Response · Authors · 2018-11-20
> > **Response to Reviewer 2 (2/2)**
> >
> > [cont] It would be fair to characterize our method as semi-amortized rather than fully-amortized as it does have the cost of a fixed number of forward-backward passes. However, our method does not require designing an inference network for a specific problem, as does Garnelo et al, etc. We retain the model agnostic property of MAML while also presenting a well-justified probabilistic inference procedure.
> >
> > => “Results from Garnelo et al are not provided for the contextual bandits experiment…”
> >
> > Please see our response to reviewer 3 with regard to not including the results from Garnelo et al.
> >
> > => “.., it is unclear how many gradient steps were taken at test time...”
> >
> > We take 5 gradient steps for training and 10 gradient steps at test time. One could argue that we should only take 5 steps at test time (as our parameterized procedure for producing an approximate posterior has been trained to produce a good approximate posterior after 5 steps). However, we found taking 10 steps to slightly improve accuracy without hurting uncertainty quantification. This is in line with the original MAML paper and (we believe) its follow-ups.
> >
> > => “The statement ‘...’ should … either be investigated or omitted...”
> >
> > We are happy to omit this from the latest revision.
> >
> > For comparison to previous work with regard to experiments, please see our response to reviewers 1 & 3. Taking the reviewers’ feedback into account, we are working on our own implementation of Probabilistic MAML to compare our model to.
> >
> > We now address the reviewer’s concerns about relation to previous work. We thank the reviewer for pointing us to work of Gordon et al. We expound more on the differences of our work compared to the previous work (and will add this to revision):
> >
> > Probabilistic MAML - does not maintain uncertainty in task-specific weights, and all uncertainty in posterior comes from uncertainty in global parameters which has fixed variance. This is inappropriate for settings where we wish to reason about task-dependent uncertainty, such as using it to guide exploration for contextual bandit problem or in few-shot learning when we desire calibrated predictive distributions.
> >
> > Bayesian MAML - has linear memory scaling in the number of particles used. This memory scaling requires one to share some parameters between particles to scale the method up to large models. Our method is more scalable in this sense and doesn’t require us to determine what parameters should be shared vs task-specific (ones we have a full distribution over) and can be discovered by the model itself. In Figure 5, we can see that the standard deviations of the prior for the 1st layer weights are very close to 0, indicating that the 1st layer is in essence shared because there will be a large penalty for changing it for a task relative to the prior. This allows more flexibility when we apply our model to new problems, as the shared vs task-specific parameter distinction can be discovered by the model itself.
> >
> > Gordon et al - use an encoder for producing the variational distribution for different tasks and also make use of parameter sharing. This again requires us to define which parameters are global, shared parameters and which parameters are task-specific and have a variational distribution over them. It also requires designing an encoder structure for each task (i.e. is not necessarily model-agnostic).
> >
> > Garnelo et al - is interesting work on achieving GP-like uncertainty quantification with NNs. However it has not yet been demonstrated whether it can be scaled to large models as required for mini-Imagenet.
> >
> > We believe the above papers consist of interesting and valuable work, but that our current work occupies an important place in the pareto frontier:
> > a. We quantify uncertainty in task-specific parameters (unlike Finn et al). We also show that this leads to well-calibrated probabilities. While Kim, Garnelo, and Gordon et al do quantify uncertainty in task-specific parameters, they do not show the impact on well-calibrated predictive distributions (and we’re not aware of other work doing this in the few-shot learning setting).
> > b. Our method is model-agnostic and requires no extra engineering of inference networks (as with Gordon et al) or careful choice of parameter sharing (as with Gordon and Kim et al). We do not need to stipulate which parameters are global and which parameters are task-specific and this is learned by the model itself based on the data. Thus, our method can be straightforwardly applied to an existing model of choice. As an example, we were easily able to apply our model to the contextual bandit setting without requiring any extra engineering of the network.
> > c. Our method is computationally efficient and scales to large problems, unlike Garnelo et al.
> > d. Our method is extendible and future work can consider more expressive variational distributions for the prior (and corresponding posterior), as has been done in VI methods for Bayesian NNs.

---

### Official Review · AnonReviewer3 · 2018-11-04
**Limited baselines in comparison, cost not clear**

**Rating:** 5
**Confidence:** 3

**Review:**

The authors proposed a meta-learning approach which amortizes hierarchical variational inference across tasks, learning an initial variational distribution such that, after a few steps of stochastic optimization with the reparametrization trick, they obtain a good task-specific approximate posterior. The optimization is performed by applying backpropagation through
gradient updates. Experiments on a contextual bandit setting and on miniImage net show how the proposed approach can outperform a baseline based on the method MAML. Although in miniImagenet the proposed method does not produce
gains in terms of accuracy, it does produce gains in terms of uncertainty estimation.

Quality:

The derivation of the proposed method is rigorous and well justified. The experiments performed show that the proposed method can result in gains. However, the comparison is only with respect to MAML and other techniques could have also be included to make it more meaningful. For example,

Gordon, Jonathan, et al. "Decision-Theoretic Meta-Learning: Versatile and
Efficient Amortization of Few-Shot Learning." arXiv preprint arXiv:1805.09921
(2018).

or the methods included in the related work section, or Garnelo et al. 2018.

The authors do not comment on the computational cost of the proposed method.

Clarity:

The paper is clearly written and easy to read.

Novelty:

The proposed method is new up to my knowledge. This is one of the first methods to do Bayesian meta-learning.

Significance:

The experimental results show that the proposed method can produce gains. However, because the authors only compare with a non-Bayesian meta-learning method (MAML), it is not clear how significant the results are. Furthermore, the computational cost of the proposed method is described well enough.

---

> ### Author Response · Authors · 2018-11-20
> **Response to Reviewer 3**
>
> Thank you for your feedback! We have addressed the two points you mentioned below.
>
> => Baselines for comparison
>
> Please see our response to Reviewer 1 with regards to comparisons to Bayesian versions of MAML for cifar and mini-Imagenet. As mentioned, we have worked on our own implementation of Probabilistic MAML and based on preliminary experiments on mini-Imagenet, it does appear that our model has better predictive uncertainty.  Our implementation does not exactly reproduce the results from the Probabilistic MAML paper and so we are in touch with the authors trying to make sure our implementation matches theirs. We hope to add the results of these experiments to the paper by the end of the revision period.
>
> For the contextual bandit experiments, even with the source code, it is not straightforward to apply the aforementioned techniques. Probabilistic MAML does not maintain uncertainty in task-specific weights - all uncertainty in the posterior comes from uncertainty in global parameters which has fixed variance. Therefore, as experience is accumulated on a new task, it is not possible to compute a posterior for the task that will become more certain. This makes Probabilistic MAML inappropriate for the contextual bandit Thompson sampling setup. Bayesian MAML could be applied to the contextual bandit experiments; however, it would likely require significant effort to tune the appropriate size of the ensemble (as Bayesian MAML maintains an approximate posterior consisting of M different copies of the model) and amount of parameter sharing so as to make experiments feasible and to have the appropriate amount of exploration. If M were too small, then Thompson sampling would provide very limited exploration. Additionally, for Gordon et al, we have a similar issue, as the amortization network they use outputs the distribution of weights over the final layer (whereas the rest of the network weights are shared) and would require tinkering to get the appropriate network architecture to work in the contextual bandits setting.
>
> We would like to stress that our method is easy to apply to the problem because we can easily sample from our model’s approximate posterior and because the total number of parameters are only increased 2-fold (weights and variances). We can calculate an approximate posterior over any commonly used network architecture for which we can compute gradients, making our method model-agnostic without introducing new hyperparameters such as the ensemble size or amount of parameter sharing which must be carefully tuned.
>
> As for Garnelo et al, we could not find details of their setup for the contextual bandit experiment (such as the network architecture, how often to update the models in each trial, how many batches to use for each update, what optimizer to use, etc), which prevented a fair comparison of inference methods. We emailed the authors several times with these questions but received no response. If the reviewers wish, we are happy to include the results from Garnelo et al in the paper, with an asterisk indicating we do not know the design of their experiment and cannot fairly compare MAML or our method to them because of the different hyperparameters we likely used.
>
> => Computation cost of method
>
> The computation cost of our method is similar to MAML except for the fact we need to compute stochastic gradients in the inner loop. To reduce the variance of the stochastic gradients, we do the following (as has been commonly done in previous work involving bayesian neural networks):
> a. Use fully-independent (or close to fully-independent) weight samples for each example in an episode.
> b. Average over multiple weight samples when computing the expectation.
>
> (a) is achieved using the local reparameterization trick for fully-connected layers and flipout for convolutional layers. Both of these methods increase the complexity of the forward pass by 2 because they require two weight multiplications (or convolutions) rather than one for normal fully-connected or convolutional layers. (b) is achieved by replicating the data. Because we are in the few-shot learning setting, we can simply replicate the episode data enough times to get different samples and average and the replicated data still fits in a forward pass on the GPU. Thus (b) doesn’t increase the time complexity too much because it corresponds to using a bigger batch of data for each episode while using the same amount of forward passes.
>
> For example, for the cifar-100 experiments, our model took 2.6 times as long to train than MAML on a single GPU. This is typical of the time tradeoff between training a bayesian vs non-bayesian deep network. We will add more details about the computational cost to a new revision.

---

### Official Review · AnonReviewer1 · 2018-11-06
**Simple and interesting, but missing some Bayesian baselines**

**Rating:** 6
**Confidence:** 4

**Review:**

The authors consider meta-learning to learn a prior over neural network weights. This is done via amortized variational inference. This means that a good initialisation of the variational parameters are learned across tasks, such that a good set of hyperparameters per task can be found in a few gradient steps. The proposed approach is evaluated on a toy and several popular benchmarks (like miniImagenet).

The topic is timely. The contribution is modest, essentially applying the same idea as the one proposed in MAML to a variational objective, but well executed. The paper is relatively well-written and the contributions clearly stated/motivated. Section 2 and 3 could be written in a more compact way (in particular the math), but it does not harm the flow. The authors conducted a good set of experiments, but are missing comparisons Bayesian versions of MAML.

---

> ### Author Response · Authors · 2018-11-20
> **Response to Reviewer 1**
>
> We appreciate your comments! With regard to comparisons with bayesian versions of MAML, unfortunately, we could not find any source code available for these models. For the cifar-100 and mini-Imagenet experiments, we cannot just take the results reported in these papers - our evaluation focuses on specific metrics used to quantify uncertainty so we would need to have the code for the models to calculate these metrics.
>
> However, taking the reviewers’ feedback into account, we have worked on our own implementation of Probabilistic MAML and based on preliminary experiments on mini-Imagenet, it does appear that our model has better predictive uncertainty (both in terms of calibration of the predictive distribution and confidence on out-of-distribution examples). Our implementation does not exactly reproduce the results from the Probabilistic MAML paper and so we are in touch with the authors trying to make sure our implementation matches theirs. We hope to add the results of these experiments to the paper by the end of the revision period.
>
> For the contextual bandit experiments, even with the source code, it is not straightforward to apply the aforementioned techniques. Probabilistic MAML does not maintain uncertainty in task-specific weights - all uncertainty in the posterior comes from uncertainty in global parameters which has fixed variance. Therefore, as experience is accumulated on a new task, it is not possible to compute a posterior for the task that will become more certain. This makes Probabilistic MAML inappropriate for the contextual bandit Thompson sampling setup. Bayesian MAML could be applied to the contextual bandit experiments; however, it would likely require significant effort to tune the appropriate size of the ensemble (as Bayesian MAML maintains an approximate posterior consisting of M different copies of the model) and amount of parameter sharing so as to make experiments feasible and to have the appropriate amount of exploration. If M were too small, then Thompson sampling would provide very limited exploration.
>
> We would like to stress that our method is easy to apply to the problem because we can easily sample from our model’s approximate posterior and because the total number of parameters are only increased 2-fold (weights and variances). We can calculate an approximate posterior over any commonly used network architecture for which we can compute gradients, making our method model-agnostic without introducing new hyperparameters such as the ensemble size or amount of parameter sharing which must be carefully tuned.

---

### Author Response · Authors · 2018-11-27
**Revision Uploaded**

We thank the reviewers again for their feedback to improve the paper. We have uploaded a new revision to address the comments we received. This revision contains the following changes:

1. Based on feedback from all 3 reviewers, we have added a comparison to Probabilistic MAML (Finn et al) for the few-shot learning experiments. Though Probabilistic MAML achieves better accuracy on the few-shot learning tasks, our model achieves better predictive uncertainty on all 3 benchmarks in terms of both reliability diagrams & associated calibration error metrics and in terms of confidence on out-of-episode examples. This is as expected, as Probabilistic MAML does not maintain a posterior over task-specific weights, instead only measuring uncertainty in the global parameters.
2. We have added more details about our exact evaluation setup for few-shot learning and re-creation of maml-based models' results in the evaluation section and added a table of hyperparameters for our model with regard to few-shot learning in the appendix.
3. We have added a description of another way to look at the final objective that we use in order to better motivate it based on feedback from reviewer 2.
4. We have updated the related work section to include citation of Gordon et al and added discussion to point out how our work differs from other Bayesian models for few-shot learning based on feedback from reviewer 2.
5. We have added a description of the computation cost of our model based on feedback from reviewer 3.
6. We had uploaded the wrong plot for the empirical CDF of entropy for the CIFAR 1-shot, 10-class benchmark in the initial submission and it has been fixed in this revision. The conclusion in terms of performance comparison remains the same in the fixed plot.

---

### Meta-Review · Area_Chair1 · 2018-12-09
**simple method for Bayesian meta-learning; paper is well-executed**

**Confidence:** 5
**Recommendation:** Accept (Poster)

**Metareview:**

This paper combines two ideas: MAML, and the hierarchical Bayesian inference approach of Amit and Meir (2018). The idea is fairly straightforward but well-motivated, and it seems to work well in practice.  The paper is well-written and includes good discussion of the relevant literature. The experiments show improvements on various tests of Bayesian inference, and include some good analysis beyond simply reporting better numbers.

On the whole, the reviewers are fairly positive about the paper. (While the numerical scores are slightly below the cutoff, the reviewers are more positive in the discussion.) The reviewers' main complaint is the lack of comparisons against recently published methods, especially Gordon et al. (2018). The lack of comparison to this paper doesn't strike me as a big problem; the preprint was released only a few months before the deadline, their approach was very different from the proposed one, and the proposed approach has some plausible advantages (simplicity, computational efficiency), so I don't think a direct comparison is required for acceptance.

Overall, I recommend acceptance.